# The Role of Mucosal-Associated Invariant T Cells in Viral Infections and Their Function in Vaccine Development

**DOI:** 10.3390/vaccines13020155

**Published:** 2025-02-02

**Authors:** Chie Sugimoto, Hiroshi Wakao

**Affiliations:** Host Defense Division, Research Center for Advanced Medical Science, Dokkyo Medical University, Mibu 321-0293, Japan; hwakao@dokkyomed.ac.jp

**Keywords:** viral infection, vaccine, innate-like T cells, mucosal-associated invariant T cells

## Abstract

Mucosal-Associated Invariant T (MAIT) cells, which bridge innate and adaptive immunity, have emerged as an important player in viral infections despite their inability to directly recognize viral antigens. This review provides a comprehensive analysis of MAIT cell responses across different viral infections, revealing consistent patterns in their behavior and function. We discuss the dynamics of MAIT cells during various viral infections, including changes in their frequency, activation status, and functional characteristics. Particular attention is given to emerging strategies for MAIT-cell-targeted vaccine development, including the use of MR1 ligands as mucosal adjuvants and the activation of MAIT cells through viral vectors and mRNA vaccines. Current knowledge of MAIT cell biology in viral infections provides promising approaches for harnessing their functions in vaccine development.

## 1. Introduction

Living organisms are constantly exposed to threats from various pathogens, particularly viruses and other foreign micro-organisms. To counter these threats, vertebrates, including humans, have developed complex immune systems through evolution. This system primarily consists of two defense mechanisms: innate immune responses and adaptive immune responses.

In the case of viral infection, the innate immune system rapidly detects viral invasion through various pattern recognition receptors (PRRs) that recognize pathogen-associated molecular patterns (PAMPs) and/or danger-associated molecular patterns (DAMPs), which activates the downstream cascades. Such an activation in turn induces antiviral cytokines, primarily type I interferons (IFN-α/β), which serve as key components of initial defense. Moreover, the innate immune response plays a crucial role in inducing and activating subsequent adaptive immune responses through the production of inflammatory cytokines and chemokines. These mediators enable the adaptive immune system to establish a virus-antigen-specific defense through antibody production by B cells and cellular immune responses by T cells. Such responses are central to vaccine development because they provide long-term immune protection against reinfection by the same pathogen through the formation of immunological memory.

In recent years, much attention has been focused on the immune cells that are distinct from the traditional dichotomy of innate and adaptive immunity, combining characteristics of both systems. Innate-like T cells represent a prime example of such populations [1]. Notably, these cells, including invariant Natural Killer T (iNKT) cells and Mucosal-Associated Invariant T (MAIT) cells, express T cell receptors (TCRs), characteristic of adaptive immune T cells, while their antigen recognition and activation mechanisms resemble those of the innate immune system. Interestingly, these innate-like T cells show marked species-specific distribution patterns: while iNKT cells are abundant in laboratory mice but relatively rare in humans, MAIT cells show the opposite pattern—they are abundant in human mucosal tissues and peripheral blood but extremely rare in laboratory mice such as C57BL/6 (Table 1) [2]. However, mice generated from MAIT-cell-derived induced pluripotent cells (MAIT-iPSCs) possessing a rearranged MAIT-cell-specific TCRα or TCRβ locus (referred to as Vα19 and Vβ8 mice, respectively) are rich in MAIT cells (Table 1) [3]. This species-specific distribution suggests that MAIT cells play a particularly important role in human immunity. Furthermore, the preferential location of MAIT cells in mucosal tissues, which serve as primary entry sites for pathogenic micro-organisms, indicates their crucial role in mounting a rapid immune response during infection.

This review focuses on MAIT cells being abundant in humans and outlines their role in viral infection. In addition, the potential applications of MAIT cells in the development of novel vaccines optimized for the human immune system will be discussed.

## 2. Characteristics of MAIT Cells as Innate-like T Cells

The semi-invariant T cell receptor (TCR) defines MAIT cells as a unique T cell subset being innate-like T cells bridging innate and adaptive immunity [6]. In humans, MAIT cells predominantly express TRAV1-2-TRAJ33/20/12 (Vα7.2-Jα33/20/12) paired with biased TRBV repertoires (predominantly TRBV6 and TRBV20), while in mice, they express TRAV1-TRAJ33 (Vα19-Jα33) predominantly paired with TRBV13 (Vβ6) and TRBV19 (Vβ8) [7,8,9]. These TCRs recognize riboflavin biosynthesis-derived metabolites, particularly 5-(2-oxopropylideneamino)-6-D-ribitylaminouracil (5-OP-RU) and its precursor 5-amino-6-D-ribitylaminouracil (5-A-RU), presented by the monomorphic MHC class I-like molecule MR1 (Figure 1) [10]. These metabolites are produced in micro-organisms, including bacteria and fungi, but not mammals, and the MR1 amino acid sequence is highly conserved across mammalian species [11,12]. These features indicate that MAIT cells play an important role in antimicrobial immunity during evolution. Following TCR-mediated antigen recognition, MAIT cells rapidly secrete inflammatory cytokines such as interferon (IFN)-γ, tumor necrosis factor (TNF)-α, and interleukin (IL)-17A, and cytolytic molecules such as granzyme B and perforin [13,14]. Complementary to this pathway, MAIT cells respond to and are activated by inflammatory cytokines, particularly IL-12 and IL-18 [15]. The latter allows MAIT cells to participate in antiviral immunity despite the absence of riboflavin metabolites in the virus (Figure 1; see Section 3). Besides their antimicrobial response, MAIT cells contribute to tissue homeostasis and repairing through IL-22 and amphiregulin, and expression of the genes relevant to tissue repairing [16,17,18].

MAIT cell development comprises the selection by MR1-expressing CD4+CD8+ double-positive (DP) thymocytes, and/or thymic epithelial cells [19,20]. The transcription factor promyelocytic leukemia zinc finger protein (PLZF), along with retinoid orphan receptor (ROR)γt and T-bet, is expressed during this process and is crucial for their rapid response and acquisition of effector functions [5,21]. After thymic egress, MAIT cells undergo peripheral expansion boosted by the commensal microbiota and environmental cues [21]. Maturation after thymic egress involves the acquisition of PLZF, followed by the expression of T-bet and RORγt, leading to functionally distinct subsets, MAIT-1 and MAIT-17, respectively [22]. Due to unique characteristics such as rapid activation and abundant localization in mucosal tissues, MAIT cells serve as important sentinels at barrier surfaces where rapid detection of pathogens is fundamental to the host defense.

Historically, MAIT cells have been identified as CD3+ and Vα7.2+ cells harboring high expression of CD161 (KLRB1) and of IL-18 receptor (R)α [13]. MAIT cells are predominantly CD8α+ or CD4-CD8- double-negative [8]. However, not all CD3+Vα7.2+CD161+IL-18Rα+ cells are authentic MAIT cells. The advent of MR1 tetramers loaded with 5-OP-RU has enabled precise identification of MAIT cells, complementing the above surrogate marker-based detection methods [9]. MAIT cells constitute approximately 1–10% of peripheral blood T cells, up to 20–45% of hepatic T cells, 2–4% of T cells in the lung, and 1–2% of T cells in the intestinal lamina propria in humans [4]. Notably, MAIT cell frequencies vary significantly among individuals and are influenced by age, sex, and disease states [23,24]

These characteristics of MAIT cells, particularly their TCR-independent activation and strategic tissue localization, have led researchers to investigate their roles in viral infections. The following section outlines recent data regarding MAIT cell dynamics during viral infections. 

## 3. MAIT Cell Dynamics in Viral Infections

The role of MAIT cells has been primarily investigated in human viral infections such as human immunodeficiency virus (HIV) [25,26,27,28,29,30,31,32,33], hepatitis C virus (HCV) [34,35,36,37,38], hepatitis B virus (HBV) [39,40,41,42], severe acute respiratory syndrome-coronavirus (SARS-CoV)-2 [43,44,45,46,47,48,49,50,51,52,53], and HIV co-infection with either HCV or *Mycobacterium tuberculosis* [54,55,56,57,58] using clinical samples from patients. Studies have also reported their involvement in infections caused by influenza viruses [59,60], dengue virus [61,62], hepatitis D virus (HDV) [63], Epstein–Barr virus [64], human T-lymphotropic virus-1 [65], hepatitis A virus [66], and hantavirus [67,68]. A comprehensive review of these studies has revealed three consistent profiles in MAIT cell responses across different viral infections, particularly in their abundance, activation states, and functional characteristics (Table 2).

One of the most striking features is a marked decrease in peripheral blood MAIT cell counts, particularly pronounced in severe cases of HIV, HCV, HBV, HDV, and SARS-CoV-2 infections. Furthermore, MAIT cell counts do not fully recover even after the virus is eliminated by antiviral therapies such as combination antiretroviral therapy (cART) for HIV [26,32] and direct-acting antiviral for HCV [36], suggesting chronic immune dysfunction.

Another consistent feature is the activation state of the residual MAIT cells. Specifically, expression of the activation markers such as CD38, HLA-DR, and CD69 is increased, concomitant with upregulation of the exhaustion markers such as programmed cell death (PD)-1, T-cell immunoglobulin mucin (TIM)-3, and cytotoxic T lymphocyte antigen (CTLA)-4 [34,40,43,55]. This phenotype most likely reflects a persistent immune activation with tissue-resident MAIT cells exhibiting a more pronounced activation and exhaustion state than those in peripheral blood [27,63].

Activation of MAIT cells during viral infection occurs primarily through a TCR-independent cytokine-mediated pathway. While in vitro studies suggest essential roles for both IL-18 and IL-12 [59,60], analysis of clinical samples indicates that IL-18 functions as a primary driver of activation, with IL-12 and type I interferons playing an auxiliary role [30,46]. This cascade is typically initiated by virus-activated monocytes and dendritic cells, which are the main sources of the inflammatory mediators even without being directly infected [59,68,69]. The importance of this pathway is underscored by the fact that MAIT cells are activated during viral infection even in the absence of riboflavin metabolites, the classical TCR ligand of MAIT cells. In vitro studies have shown that TCR- and cytokine-activated MAIT cells exhibit different but similar transcriptional profiles [16,17,18]. Notably, virus-activated MAIT cells are likely to have impaired responses to bacterial infection, as evidenced by their weakened production of inflammatory cytokines such as IFN-γ and TNF-α in response to TCR/MR1-dependent stimuli [26,36,37,45].

Multiple mechanisms contribute to the reduction in peripheral blood MAIT cell numbers during viral infections. Direct evidence of tissue accumulation has been demonstrated in COVID-19 patients, where MAIT cells are enriched in the airways [43] and lung tissue [53]. MAIT cells constitutively express tissue homing molecules including chemokine receptors CCR5 and CCR6, and the integrin CD103 [54,55]. During viral infections, while the tissue retention marker CD69 is consistently upregulated [25,26,28,32,34,35,39,42,43,44,45,46,49,55,59,61,63,70], changes in other homing markers show more complex patterns that vary by infection type and disease stage. Indeed, studies in SIV infection have shown concurrent depletion of MAIT cells across multiple anatomical sites [70], suggesting that mechanisms beyond tissue redistribution are involved. The decrease in or loss of MAIT cells is also caused by interrelated processes of cell death and exhaustion. Persistent exposure to inflammatory signals leads to increases in inhibitory receptors like PD-1 [31,34,36,40,42,52,55,57,63]. This state of cellular exhaustion not only impairs MAIT cell function but also increases susceptibility to cell death. Both apoptotic and pyroptotic cell death pathways are involved, and inflammatory cytokines further promote these processes [30,49,51,63]. Mutual reinforcement between exhaustion and cell death may explain the persistent depletion of MAIT cells observed even after viral clearance [26,36]. However, the precise molecular mechanisms underlying these functional changes remain to be elucidated. A comprehensive analysis of the transcriptional landscape will reveal how inflammatory signals affect MAIT cell function during viral infection.

## 4. Significance of MAIT Cell Dynamics in Viral Pathogenesis

Through the production of inflammatory cytokines and cytolytic molecules, the activation of MAIT cells has two opposite consequences: enhancing protective immunity or exacerbating disease.

In HIV infection, the pathophysiological significance of MAIT cells is largely related to intestinal immune dysfunction. Depletion of intestinal CD4+ T cells in early HIV infection leads to mucosal barrier dysfunction, which in turn engenders subsequent microbial translocation, triggering systemic immune activation [71]. Such persistent stimulation makes MAIT cells dysfunctional. Since MAIT cells are activated by bacterial vitamin B2 metabolites and play an important role in antimicrobial defense, their dysfunction or depletion likely contributes to increased susceptibility to opportunistic infections in HIV-infected individuals [25,26,28,54]. Notably, MAIT cell function is not fully restored even after the viral load is controlled and CD4+ T cell numbers are recovered upon cART [25,26,28,29,54]. This serves as an important indicator of incomplete immune reconstitution in HIV infection and explains the persistent risk of bacterial opportunistic infections despite successful viral suppression.

MAIT cells, which comprise 20–50% of liver T cells, could play crucial roles in chronic viral hepatitis caused by HCV, HBV, and HDV. Studies have consistently reported reductions in MAIT cell numbers in both peripheral blood and the liver, and a characteristic of residual MAIT cells in the liver is their highly activated phenotype (expressing CD69, CD38, and HLA-DR) [34,36,37,38,40,41,42,60,63]. These cells also show an enhanced cytolytic capacity, as evidenced by increased expression of the degranulation marker CD107a [34,39,63]. These features suggest that MAIT cells may instigate tissue damage in chronic inflammatory conditions. Indeed, the degree of MAIT cell reduction correlates with the severity of hepatitis inflammation and fibrosis [36,37]. Furthermore, MAIT cell dysfunction may be associated with an increased susceptibility to bacterial infection in patients with chronic liver disease [28,34,63].

In acute viral infections, MAIT cells may play dual roles in immune responses. During the early stage of infection with influenza viruses or dengue virus, MAIT cells are activated primarily through IL-18 and type I interferons, contributing to antiviral responses through IFN-γ and granzyme B production [59,60]. Interestingly, in H7N9 influenza virus infection, high MAIT cell counts have been reported in peripheral blood mononuclear cells (PBMC) of the recovered patients, suggesting a protective role [59]. Similarly, in mouse models of influenza virus infection, the protective role of MAIT cells has been demonstrated. Specifically, MR1 knockout (MAIT-cell-deficient) mice exhibit enhanced weight loss and mortality upon influenza virus challenge, which is mitigated by an adoptive transfer of MAIT cells [35]. Conversely, in COVID-19 and dengue fever, MAIT cell activation is more pronounced in severe patients, suggesting their implication in disease exacerbation through excessive immune responses [43,61]. Particularly in COVID-19, marked peripheral blood reduction of MAIT cells concomitant with selective accumulation in the lung is prominent in severe cases, indicating exacerbated lung tissue damage [46,50]. Although MAIT cells show transient recovery after severe COVID-19, they exhibit persistent functional impairment beyond 9 months, characterized by elevated PD-1 expression and reduced polyfunctionality, which may be an immunological basis for long COVID pathophysiology [52].

## 5. Therapeutic Targeting of MAIT Cells in Viral Infections

Despite the growing interest in MAIT-cell-based therapies, evidence from in vivo therapeutic interventions remains remarkably limited. Current therapeutic strategies targeting MAIT cells primarily focus on cytokine-based approaches and receptor blockade (Table 3). The most extensively studied approach involves IL-7 treatment, which is known to enhance T cell proliferation and survival. Initial mechanistic insights came from an in vitro study. IL-7 was shown to effectively ’arm’ MAIT cells by enhancing their cytolytic capacity and cytokine production. Additionally, it could restore the function of impaired MAIT cells from HIV-1-infected individuals [29]. A clinical study involving HIV-infected patients has shown that IL-7 treatment increased both relative and absolute numbers of MAIT cells, with expansion occurring predominantly in the CD8+ subset. This restoration of MAIT cells suggests potential for immune reconstitution in chronic viral infections [33]. Similarly, a study with COVID-19 patient samples showed IL-7 treatment partially restored MAIT cell functionality by enhancing perforin expression [48].

Combinatorial cytokine approaches have also been explored in a macaque SHIV infection study [72]. The combination of IL-12, IL-18, and IL-7 demonstrated synergistic effects, enhancing both the proliferation and cytolytic function of MAIT cells. This multi-cytokine approach may be particularly valuable in cases where single cytokine treatments show limited efficacy.

An alternative ex vivo study using cells from HIV-infected individuals demonstrated that anti-IL-10 receptor blockade improved MAIT cell responses to opportunistic infections such as *Mycobacterium tuberculosis* and restored MAIT cell dysfunction. These findings highlight the potential importance of targeting immunoregulatory pathways to enhance MAIT cell function.

While these results are promising, further investigation is required for clinical translation, particularly regarding the tissue-specific effects of these treatments in key sites of viral infection such as the mucosa and liver. Future therapeutic strategies may need to take into account the complex interactions between MAIT cells and other immune cell populations and the unique circumstances of various viral infections. Combined approaches targeting multiple aspects of MAIT cell biology may ultimately reveal the most effective way to restore protective immunity in viral infections.

## 6. Strategies for Development of MAIT-Cell-Targeted Vaccines

MAIT cells have distinct properties that make them particularly attractive for enhancing vaccine-induced immunity; their rapid cytokine production and interaction with antigen-presenting cells enable them to serve as a crucial orchestrator between innate and adaptive immune responses [74]. This bridging function, coupled with its preferential localization in the mucosa, provides an attractive opportunity for prophylactic vaccine development. Recent studies have revealed some approaches to harness MAIT cells for enhancing vaccine efficacy (Figure 2).

While conventional protein antigens are unable to activate MAIT cells, combining them with MAIT cell agonists offers a promising vaccination strategy. In this context, a study targeting the TCR-MR1 axis evaluated the efficacy of 5-OP-RU (a MAIT cell cognate antigen) as a mucosal adjuvant in mice (Figure 2A) [75]. The results show enhanced vaccine responses upon administration of 5-OP-RU and either SARS-CoV-2 spike protein or influenza virus hemagglutinin. Activation mediated through CD40-CD40L interactions between MAIT cells and dendritic cells promotes T follicular helper cell differentiation and thereby induces robust humoral immunity. This approach succeeds in provoking production of systemic IgG and mucosal IgA antibodies while minimizing inflammatory responses compared to conventional vaccine adjuvants. Notably, while intranasal administration of 5-OP-RU alone provides partial protection against lethal influenza virus challenge, co-administration with the viral antigens leads to further protection [75].

Studies on viral vectors and mRNA vaccines revealed that these platforms activate MAIT cells via the cytokine-dependent pathway; viral vectors activate them through their infection cycle, while mRNA vaccines through the recognition of foreign RNA as PAMPs (Figure 2B). These processes trigger innate immune responses, resulting in cytokine production and subsequent MAIT cell activation. For instance, non-replicating chimpanzee adenoviral vector (ChAdOx1) vaccines result in the production of IFN-α from the plasmacytoid dendritic cells and of IL-18 from the monocytes [76]. IFN-α in turn stimulates monocytes to produce TNF-α and these three cytokines are required for MAIT cell activation. In ChAdOx1-COVID19 vaccines, the degree of MAIT cell activation correlates with the induction of antigen-specific T cells. The essential role of MAIT cells in ChAdOx1 vaccine responses was confirmed in experiments using MR1 knockout mice, which show markedly attenuated SARS-CoV-2 spike-specific CD8+ T cell responses following vaccination in the absence of MAIT cells [76]. The role of vector-activated MAIT cells in enhancing vaccine responses was also observed in a macaque prime-boost vaccine study for the AIDS envelope protein vaccine, where a replicating adenoviral vector induced MAIT cells to enhance secretion of the cytokines and chemokines that regulate B cell responses [77]. In the mRNA vaccine platform, SARS-CoV-2 mRNA vaccination induces the polyclonal expansion of MAIT cells with enhanced TNF-α production, thus promoting B cell responses in a manner dependent on TNF-α [78]. However, the relationship between MAIT cell activation and mRNA vaccine efficacy appears to be more complex; MAIT cell counts in individuals positively correlate with SARS-CoV-2-specific immune responses, whereas excessive activation correlates negatively with resulting immune responses [80]. The findings suggest that MAIT cell activation through the different vaccine platforms needs to be carefully controlled, as the level of activation influences vaccine efficacy either positively or negatively.

Recent studies have reported an innovative approach combining the MR1 ligand 5-OP-RU with replication-competent viruses, which use two distinct activation pathways: a direct MR1-dependent and a cytokine-dependent activation, triggered by 5-OP-RU and type I interferons, respectively [79]. The latter is produced through Toll-like receptor 3 signaling induced by viral double-stranded RNA in monocytes/macrophages/dendritic cells (Figure 2C). This parallel activation resulted in MAIT cell accumulation in afflicted tissues, differentiation into inflammatory MAIT-1, and enhanced virus-specific CD8+ T cell responses, ultimately providing improved protection against influenza viruses. It is notable that repeated administration of 5-OP-RU does not induce MAIT cell anergy, suggesting its potential utility in prime-boost vaccination protocols [79].

## 7. Future Direction of MAIT Cell Research in Viral Immunity and Vaccine Design

Natural viral infection results in a decrease in the number of MAIT cells, increased expression of activation markers, and dysfunction. In chronic infections such as AIDS, viral hepatitis, and COVID-19, the degree of these MAIT cell alterations correlates with disease severity. It remains, however, challenging to distinguish whether MAIT cell activation is a cause of disease progression or a consequence of persistent inflammatory conditions, as chronic inflammation is fundamental to these pathologies. The observation that disease progression accelerates when MAIT cells are depleted or dysfunctional might suggest a protective role for MAIT cells under normal conditions. Studies on early-stage infection offer the opportunity to elucidate the intrinsic properties of MAIT cells. Our findings in AIDS research have demonstrated that early immune responses determine subsequent infection outcomes [81,82,83]. In particular, IL-15-responsive CD8α+ cells, which may share functional characteristics with MAIT cells, are critical in determining successful viral control through the strength of their early response [84]. Given that MAIT cells bridge innate and adaptive immunity, the early stage of infection represents the most critical phase for elucidating their essential functions. Evidence supporting the protective role of MAIT cells under well-controlled infection has accumulated, including higher MAIT cell numbers in recovery cases of H7N9 influenza infection and the exacerbation of infection in MR1 knockout mice [35,59,76]. Furthermore, viral vector vaccines exhibit MAIT cell activation profiles that contribute to protective immunity [76,77].

On the other hand, MAIT cell activation in vaccine platforms enhances antigen-specific immune induction. However, studies on mRNA vaccines suggest that excessive MAIT cell activation might compromise induction of the vaccine antigen-specific antibody, highlighting the need for precise functional tuning of MAIT cells. Viral vector and mRNA vaccines, however, inherently activate the innate immune system and induce cytokine release, which makes the fine-tuning of MAIT cell activation challenging. Thus, systematic evaluation using PRR ligands and conventional adjuvants as innate immune activators may help assess how activation timing, duration, and intensity influence immune induction [79,85]. Alternatively, combining viral vector or mRNA vaccines with TCR-MR1-dependent stimuli such as 5-OP-RU could better regulate MAIT cell activation [79]. Detailed analysis of such platform-specific MAIT cell activation could lead to novel strategies for improving vaccine efficacy.

The development of appropriate mouse models is also essential for advancing these studies. While MR1 knockout mice have revealed unique immunological functions of MAIT cells, conventional laboratory mice have limited usefulness due to their low MAIT cell numbers. Our approach generates novel mice referred as Vα19 and Vβ8 mice, with MAIT cell frequencies comparable to or superior to those in humans and proper tissue-specific distribution of the functional subsets MAIT-1 (T-bet+) in the spleen and MAIT-17 (RORγt+) in the lung through chimeric mice produced from MAIT cell-iPSCs (Table 1) [3,86]. These mice have demonstrated that MAIT cells alleviate airway inflammation through group 2 innate lymphocyte suppression and inhibit cancer metastasis via enhanced cytolytic and NK cell activity [3,86,87]. Although these models may not fully mimic human infections, they provide valuable insights into early mucosal immune responses, particularly the crosstalk between MAIT cells and other immune cells during the development of adaptive immunity. The integration of insights from both basic and applied research will pave the way for more effective and safer vaccine development and infection control strategies.

## 8. Conclusions

MAIT cells, the most abundant innate-like T cells in humans, bridge innate and adaptive immunity. Given that MAIT cells are activated through TCRs and cytokines, further exploring their immunological function in bacterial and viral infections is promising for developing new preventive and therapeutic strategies. The next decade will usher in a new era in which detailed mechanistic insights into how MAIT cells, along with other immune cells, exert their antiviral functions will pave the way for the development of novel vaccines with enhanced efficacy.

## Figures and Tables

**Figure 1 vaccines-13-00155-f001:**
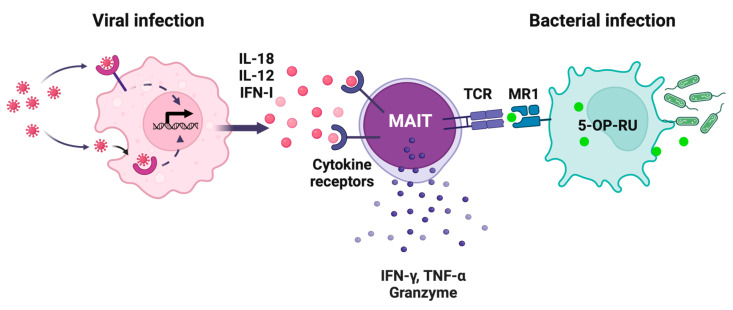
Distinct pathways in MAIT cell activation during viral and bacterial infections. MAIT cells are activated through two distinct pathways: a cytokine-mediated pathway during viral infection and a TCR-dependent pathway during bacterial infection. In viral infection, virus-infected cells or APCs recognize viral components through pattern recognition receptors (PRRs) such as TLRs, triggering the production of IL-12 and IL-18. Cytokine binding to the cognate receptors on MAIT cells initiates signal transduction, resulting in the production of IFN-γ, TNF-α, and granzymes in MAIT cells. In bacterial infections, the metabolite 5-OP-RU from the riboflavin biosynthesis pathway is presented on MR1 molecules, and recognized by the MAIT cell TCR, which in turn triggers the activation. Created in BioRender. Sugimoto, C. (2025) https://BioRender.com/p37q702 (accessed on 01 February 2025).

**Figure 2 vaccines-13-00155-f002:**
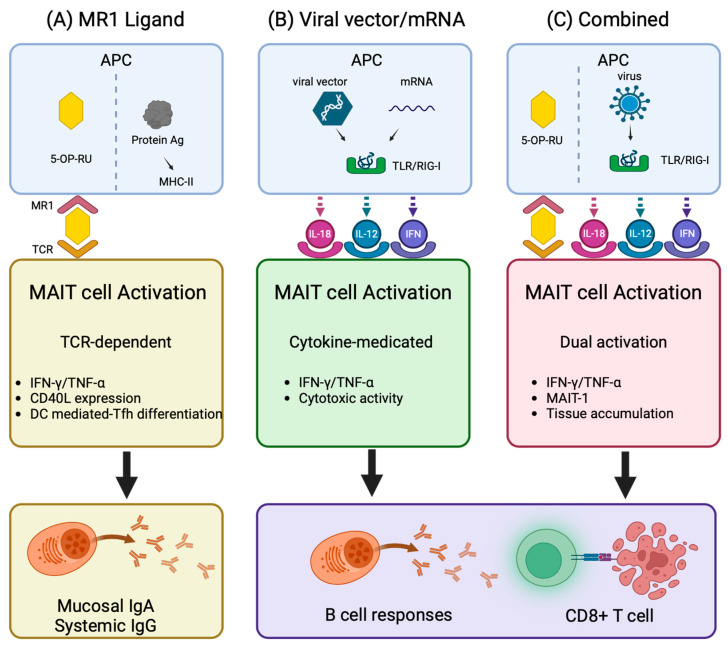
Summary of MAIT-cell-targeted vaccine strategies. (**A**) MR1 ligand strategy: Co-administration of 5-OP-RU and protein antigens activates MAIT cells through TCR-mediated recognition of MR1 ligands. Activated MAIT cells produce inflammatory cytokines (IFN-γ, TNF) and enhance B cell responses through Tfh differentiation via CD40/CD40L-mediated interactions with dendritic cells. This enhanced B cell activity promotes production of protein antigen-specific mucosal IgA and systemic IgG [75]. (**B**) Viral vectors/mRNA strategy: Viral vectors and mRNA vaccines activate MAIT cells through cytokine-dependent pathways mediated by APC-derived IL-18, IL-12, and type I IFN. Although these platforms share common activation mechanisms, they elicit distinct immunological outcomes: non-replicating adenoviral vectors (ChAdOx) enhance antigen specific CD8+ T cell responses [76], while replication-competent adenoviral vectors promote B cell responses in non-human primate models [77]. mRNA vaccines enhance B cell responses [78]. (**C**) Combined strategy: Combination of 5-OP-RU and replicating virus engages both TCR- and cytokine-elicited pathways. This dual activation promotes MAIT-1 differentiation, leading to IFN-γ/TNF production, and tissue accumulation, thereby enhancing antigen-specific CD8+ T cell responses [79]. Created in BioRender. Sugimoto, C. (2025) https://BioRender.com/z83u834 (accessed on 1 February 2025).

**Table 1 vaccines-13-00155-t001:** MAIT cell frequencies in humans, wild type mice (C57BL/6), and MAIT-cell-rich mice (Vα19 and Vβ8 mice) in different tissues ^1^.

Tissues	Humans ^2^	C57BL/6 ^3^	Vα19 Mice ^4^	Vβ8 Mice ^4^
Peripheral blood	1–10%	<0.1%	20–30%	0.5–1%
Lung	2–4%	0.5–3%	30–35%	1–3%
Liver	20–45%	0.2–1%	35–40%	0.5–2%
Spleen	<1%	<0.1%	20–25%	1.5–2%
Intestinal lamina propria	1.5–4%	1%	15–25%	1–4%

^1^ MAIT cell frequency is shown as a percentage (%) relative to CD3+ T cells in the indicated tissue. ^2^ Kurioka et al. [4]. ^3^ Rahimpour et al. [5]. ^4^ Sugimoto et al. [3].

**Table 2 vaccines-13-00155-t002:** MAIT cell dynamics in representative viral infections *.

Viral Infections	Peripheral Blood			Affected Tissues			
	Frequency	Phenotype	Function	Tissues	Frequency	Phenotype	Function
HBV	Variable (non-significant to severely decreased)	Increased CD38/HLA-DR/PD-1	Preserved cytokine production	Liver	Maintained	Increased CD69	Preserved cytokine production
HCV	Decreased	Increased CD38/HLA-DR/PD-1	Impaired cytokine production	Liver	Decreased	Increased CD69	Impaired cytokine production
HIV	Severely decreased	Increased CD38/HLA-DR/PD-1	Impaired cytokine production	Rectal Mucosa	Maintained	Increased CD69	Preserved cytokine production
				Bronchoalveolar	Decreased	Increased PD-1/TIM-3	Impaired cytokine production
				Lymph Nodes	Decreased	Increased CD38/HLA-DR	Not reported
SARS-CoV2	Severely decreased	Increased CD38/HLA-DR/PD-1	Enhanced cytokine production	Lung	Increased	Increased CD69	Enhanced cytokine production

* The table is based on data from multiple studies, thus findings may vary.

**Table 3 vaccines-13-00155-t003:** Therapeutic and prophylactic strategies targeting MAIT cells.

Strategies	Viral Infections	Key Effects	Refs.
IL-7 treatment	HIV, COVID-19	MAIT cell restoration and functional enhancement	[29,33,48]
Cytokine combination (IL-7/12/18/)	SHIV	Enhanced proliferation and cytolytic function	[72]
Anti-IL-10R blockade	HIV	Improved anti-microbial responses	[73]

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
