# Peer review of "The Role of Mucosal-Associated Invariant T Cells in Viral Infections and Their Function in Vaccine Development"

_vaccines, 2025, doi:10.3390/vaccines13020155_

Round 1
Reviewer 1 Report
Comments and Suggestions for Authors
This manuscript provides a nice overview of the current understanding of MAIT cells and their role in defense against viral and bacterial pathogens and in the development of immunopathological conditions in infections. The authors have summarized very well the known data on the development of these cells, their phenotype, and their fate in various viral infections. An important idea of the authors is that existing antiviral vaccines can be improved by targeted stimulation of MAIT cells, as they can serve as a crucial orchestrator between innate and adaptive immune responses. Such mucosal vaccines could be a new class of vaccines that protect against respiratory pathogens at the point of their entry.
The paper is well-written and presents this complex analysis in a concise form. I have only one suggestion that could further improve this manuscript and allow the authors to better understand the material presented. Specifically, I recommend that the authors add a summarizing table with dynamics of MAIT cells during different viral infections in different anatomical signs. Particularly, it is interesting to know whether there is a different between viruses infecting respiratory tissues and viruses targeting intestine.
Also, please be consistent in spelling the term Mucosal-associated invariant T cells (e.g. a key word is “mucosal associated-invariant T cells”)
Reviewer 2 Report
Comments and Suggestions for Authors
Sugimoto et al., have submitted the review entitled “The Role of Mucosal-Associated Invariant T cells in Viral Infections and Their Function in Vaccine Development”.
This is an interesting and unique review of MAIT cells. The authors have mostly elaborated on location, function, and therapeutic strategies considering the MAIT cells.
The review is well captured with two illustrations.
This reviewer has the following suggestions.
· Lines 72-71: Could you please elaborate on the detailed mechanism of how the MAIT cells participate in antiviral immunity despite the absence of riboflavin metabolites in the virus?
· I suggest authors generate to tables containing the location and percentage of the MAIT cells in the mice and humans. Also, one more table provides the details of studies focused on prophylactic and therapeutic strategies with MAIT cells.
